# Model Test Analysis of Groundwater Level Fluctuations on Karst Cover Deformation Taking the Monolithic Structure of Guilin as an Example

Xuejun Chen [1,2], Xiaotong Gao [1,2], Hui Li [1,2], Mingming Xue [1,2], Xiaohui Gan [1,2] and Yu Song [1,*]

1   College of Civil and Architecture Engineering, Guilin University of Technology, Guilin 541004, China
2   Guangxi Key Laboratory of Geomechanics and Geotechnical Engineering, Guilin 541004, China
*   Correspondence: songyu119@glut.edu.cn

**Abstract:** Engineering practice and real-life cases show that the geological conditions of the Guilin overlying karst site are complex. In particular, the groundwater, which drives the accelerated formation of soil cavities, and the thickness of the overlying soil layer, which affects the speed of the groundwater subsidence process. Therefore, this paper is based on a physical model to evaluate the effects of groundwater level changes caused by different factors on the deformation of karst cover. The model tests are simulated for different cover thicknesses (6 cm, 9 cm, 12 cm, 15 cm, 18 cm) under rainfall and other recharge, cavity supply, and drainage conditions at the same density ($1.40\ \mathrm{g/cm^3}$) and initial water content (30%), respectively. The results show that with the increase of rainfall and other recharge time, the basic change trend of different cover thicknesses is that the infiltration curve changes faster at the beginning and slows down at the end, but the thicker the cover, the slower the overall deformation; at a certain rate of cavity recharge and drainage, the thicker the cover, the smaller the deformation caused by the fluctuation of groundwater level. The cavity recharge makes the cover displacement obvious, in the order of 0.304 cm, 0.173 cm, 0.118 cm, 0.068 cm, and 0.056 cm. After the formation of the cavity, the rainfall, other recharge, and the cavity supply and drainage accelerate the destruction and deformation of the soil body and the upward development of the cavity. The research results provide theoretical support for the subsequent prevention and control of karst collapse in covered karst areas, and have certain practical engineering significance.

**Keywords:** karst collapse; physical model; groundwater level changes; cover deformation





## 1. Introduction

Karst collapse is a karst-driven geological action and phenomenon in which the rock and soil above a karst cavity are damaged by deformation caused by natural or man-made factors, and a collapse pit (hole) is formed on the ground [1]. Karst collapse is a widespread geohazard problem worldwide, with more than 1/3 of China's land area covered by soluble rock [2]. The widespread distribution and sudden hazard of karst collapse have attracted the great attention of the international community. In the past two decades, the problem of karst collapse has become more and more serious, especially in the present day when roads are opened on every mountain and bridges are built over every river; some underground projects can hardly avoid crossing the karst development area, leading to the accelerated generation of karst collapse [3–5]. This geological disaster problem has seriously hindered local economic construction and social development and has posed a certain threat to the safety of people's lives and properties [6]. Therefore, scholars at home and abroad have paid great attention to karst collapse, and have carried out a lot of scientific research work and achieved fruitful scientific results. At present, the research results on karst collapse at home and abroad can be summarized as follows: collapse causes and collapse mechanisms, influencing factors, stability analysis and prediction, physical model tests and numerical research, monitoring, and prevention [7–11].

Karst collapse in China is mainly influenced by a variety of factors such as topography, stratigraphic lithology, geological formations, changes in groundwater dynamics, and human engineering-economic activities [12,13]. Field tests are limited in many ways, making it difficult to map the characteristics of the overlying karst soil cavities and the corresponding collapse conditions, and even more difficult to determine a series of standard test conditions that are fully compatible with them. Among the many triggering factors affecting karst ground collapse, groundwater is the most important one. The changes in groundwater level, magnitude, and frequency, and its hydrodynamic characteristics are the most active and dominant dynamic factors in the formation of karst collapse [14–18]. The vast majority of karst collapses occur under dramatic changes in groundwater dynamic conditions, and the causes are categorized by domestic scholars as subduction, disintegration, vacuum absorption, and airburst [19].

In the karst collapse of groundwater changes in the collapse mechanism, some scholars have studied many cases of karst ground collapse closely related to the changes in the water table, analysis of the impact of groundwater on karst ground collapse, verification of the "subduction theory, vacuum absorption theory, airburst theory" of the three existing mechanisms [20,21], and it is also pointed out that the types of collapse and the collapse mechanism are different under different geological conditions. The number of karst collapse hazards caused by fluctuating water levels is also increasing, exacerbating the threat and damage to life and property [22]. Therefore, for the collapse induced by the change in groundwater level, the study of the development law of cover deformation due to groundwater fluctuation is particularly important for karst collapse, and it is of great value for the prediction and prevention of karst collapse. However, as karst collapse is characterized by suddenness, concealment, complexity and uncertainty [23–26], a comprehensive and systematic theory of collapse has not yet been established. Other scholars have carried out research work on the causes and influencing factors of karst collapse by taking the final state of karst collapse as the research object [27]. It is difficult to understand the process of karst collapse, which leads to an inability to fully and deeply cognize the mechanism of collapse formation [28], so it is necessary that the formation process and evolution mechanism of karst collapse can be dynamically observed with the help of indoor physical model tests.

At the same time, indoor model tests are also one of the effective methods to verify the existence of the proposed collapse theory [29]. For example, Jin Honghua et al. [30] established a cylindrical soil collapse model and derived the calculation method of stability coefficients for four different working conditions based on the limit equilibrium theory. Sun Jinhui [31] carried out physical model tests on karst collapse with clay overburden and obtained a critical collapse width of 2–3 mm. However, the model device collapse hole was in contact with the atmosphere and in an unpressurized state, which did not match the actual karst collapse opening or collapse pipe in a pressurized state. Li Caihua et al. [32] established a physical model based on the exploration data of a collapse site, and conducted a series of tests mainly for the binary structure (clay in the upper layer and sandy soil in the lower layer). It was concluded that fluctuations in the karst water table were the main cause of the induced karst collapse, but the process of collapse formation could not be quantified. Although the use of physical model tests also verified the validity of some of the collapse theories and reproduced the process of karst collapse, which improved the understanding of the evolution of the karst collapse mechanism, there are still many questions on how to quantify the process of karst collapse, especially how to obtain the extent of cover collapse during the collapse process. Therefore, in this paper, clayey soils are selected as the cover layer, and physical model tests are carried out to simulate the collapse process by vacuum suction and collapse process by positive pressure airburst, and to simulate the cover deformation at the same density and supply and drainage rate for five different cover thicknesses respectively.

Therefore, this paper takes the one-dimensional structural overburden in the karst collapse-prone area of Lingui District, Guilin City as the research object. Based on the

previous research results, a physical model of the overburden type is constructed, a typical test model of groundwater level fluctuation is established and model tests are carried out. In this study, the effects of rainfall and other recharge and cavity supply and drainage on the settlement process of the overlying karst cover are simulated. The effect of groundwater level fluctuations on the cover deformation was then analyzed by means of the cover displacement. This study provides a theoretical and technical basis for practical engineering activities in covered karst areas.

## 2. Geological Setting of the Study Area

### 2.1. Meteorology and Hydrology

The study area is located in Lingui District, Guilin, at the southern end of the Xianggui Corridor, at an average altitude of 150 m. Meteorological observations show that the annual mean temperature in the study area ranges from 17.0 to 20.0 °C, with cold winters, the average daily temperature in January and February remaining at 5 to 7 °C, and the annual extreme minimum temperature at −5 to 0 °C. The summer and autumn seasons are significantly warmer, with extreme maximum temperatures of 37.0 to 39.0 °C. Temperatures are significantly higher in summer and autumn, with annual maximum temperatures ranging from 37.0 to 39.0 °C. The total annual precipitation is 1266.0~1986.0 mm, and the spatial and temporal distribution of rainfall is uneven, with more precipitation in spring and summer, and severe droughts in autumn and winter, with heavy rainfall mainly concentrated in May–June. The water level in the creek is 154.60 m. The water level in the creek is affected by the seasons and is mainly recharged by rainfall and domestic wastewater from the surrounding area, with a small volume of water. The water table in the study area is basically located at the bedrock surface and is subject to seasonal variations, with a range of 2–3 m.

### 2.2. Geological Features

The geological drilling data available in the study area shows that the Quaternary strata in the study area are widely distributed, mainly consisting of red clay formed by the Upper Pleistocene Alluvium ($Q_3^{al-pl}$), the Upper Pleistocene Residual Slope Formation ($Q_3^{dl-el}$) and the Upper Devonian Rongxian Formation ($D_3r$). The Upper Pleistocene Alluvium ($Q_3^{al-pl}$) is a yellow and tawny pebble-gravel sandy clay and chalky clay, composed of gravel, clay and a small amount of fine sand and pebbles, 2–14 m thick; the Upper Pleistocene Residual Slope Formation ($Q_3^{dl-el}$) is a brick-red and light yellow clay, sandy clay and sandy-clay gravel layer, containing a small amount of quartz particles, 0.1–10 m thick; the Upper Devonian Rongxian Formation ($D_3r$) is greyish-white, slightly weathered, cryptocrystalline in structure, medium-thick laminated. The main mineral is calcite, with calcite veins developed and tightly cemented, the rock is relatively intact, the core is columnar, the bedrock surface within the site is highly undulating, and the karst fissures are relatively developed (Figure 1).

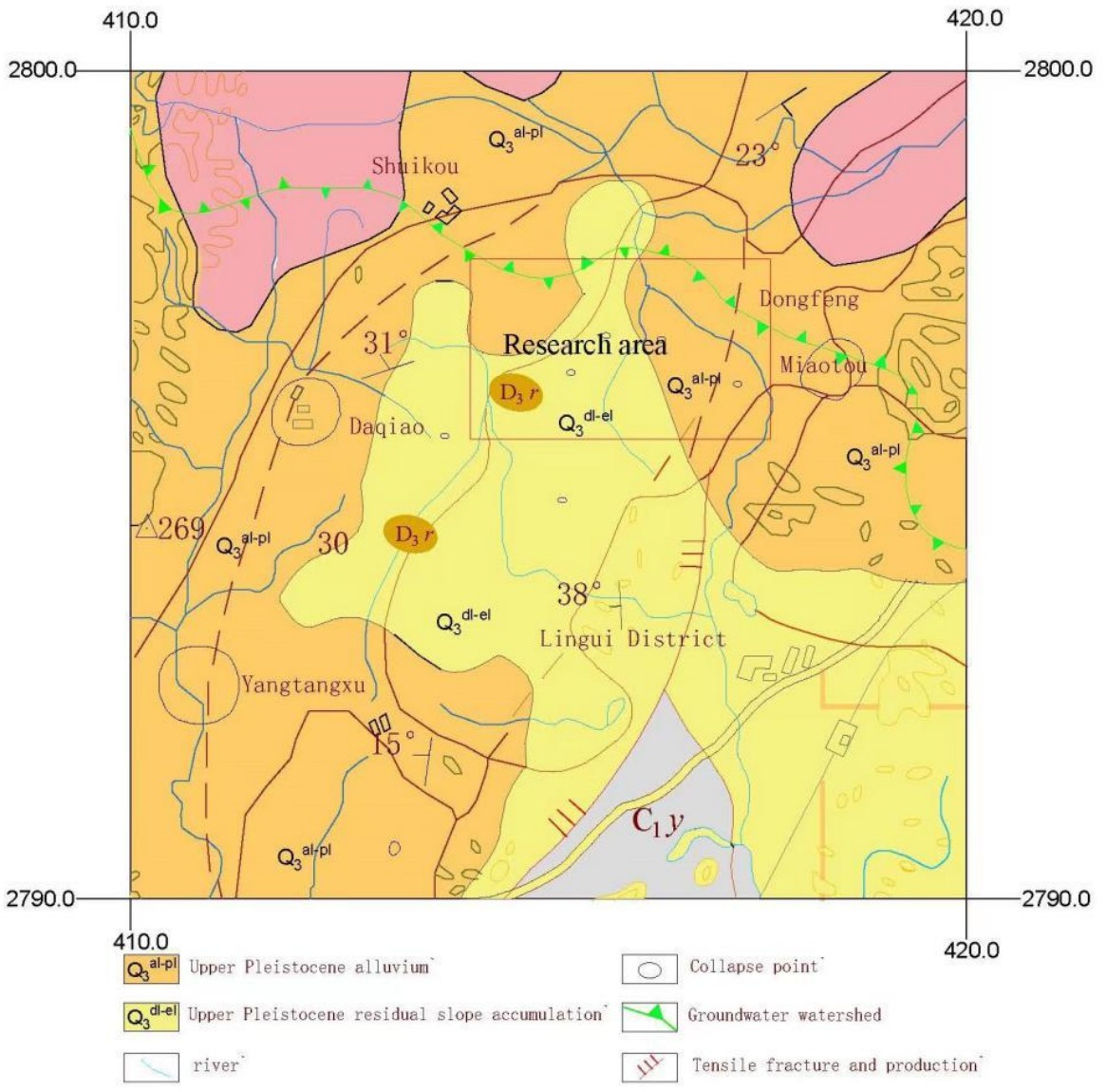

**Figure 1.** Engineering geological map.

## 3. Test Materials and Methods

### 3.1. Generalisation of the Geological Model of the Study Area

Due to the complex geological conditions in the study area, there are many uncertainties. If the karst-induced collapse test were to be carried out directly at the study site, it would be constrained by many factors. It is therefore difficult to carry out in-situ karst collapse tests at the site. In order to obtain the results of the karst collapse test under changing groundwater levels and to carry out the test successfully, a geological model of the geological conditions in the study area needs to be generalised here.

The geological drilling data collected from Lingui District, Guilin City, shows that the overlying karst in Lingui District can be mainly classified into single or double-layer structures. Among the monolayer structures, the overlying soil layer is clayey. In this simulation test study, a specific study area was selected to focus on the monolithic structure (Figure 2).

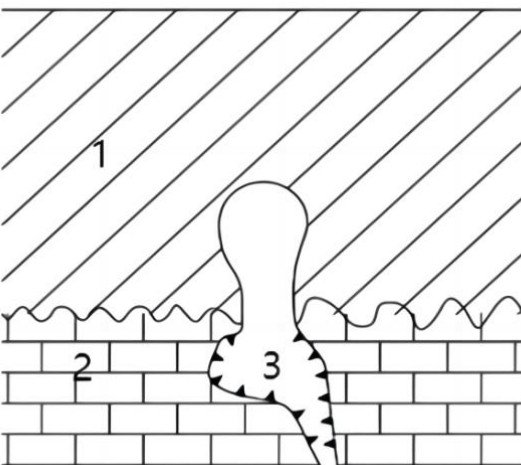

**Figure 2.** Geological generalization of a single structure. 1—Overburden clayey soil; 2—Limestone; 3—Caves.

### 3.2. Physical Modelling Principles

When conducting physical model tests, not only should the test requirements and operability be met, but similar issues should also be considered. According to the geological reports available in the study area, it is known that the thickness of the collapse prone overburden in the study area ranges from 0.7 to 5.6 m, and the diameter of the karst openings on the bedrock surface varies from 0.5 to 1.2 m. The geometric similarity ratio was finally determined to be 1:15 by combining the geological, hydrological and tectonic conditions of the study area. 6 cm diameter of the reserved soil hole was set, and the thickness of the overburden layer was 6 cm, 9 cm, 12 cm, 15 cm, and 18 cm. The soil samples of the overburden layer were taken directly from the site during the test, and the moisture content of the test was close to that of the site. However, the density was difficult to achieve in situ, and after repeated indoor tests, the density was finally determined to be 1.4 g/cm$^3$. In order to make the simulation conditions as close to the site as possible, to take into account the factors involved in karst collapse at the site and to ensure that the soil simulation parameters are basically the same as the original soil at the site, the karst collapse test device was specially tailored. In the course of the test, the test device is subjected to a large load when filling a predetermined thickness of overlying soil. This is why the frame of the device is made of a high hardness aluminum alloy and the side walls are inlaid with 5 mm thick toughened glass. To accommodate the inflow of lateral recharge water during the test, holes were drilled in the glass on the left and right sides, as shown in Figure 3a. To prevent loss of cover soil in the small holes on the left and right sides, towels were taped to the left and right sides before the test began to fill with soil, as shown in Figure 3b. In order to have a clear view of the deformation and damage process inside the cover soil, the karst access opening was set up in a semi-circle at the midpoint of the front side. A special ice block was pre-buried above the passage opening to form a pre-existing soil cavity in the shape of a 1/4 sphere during the test. The physical model consists of four parts: the main model, the rainfall system, the water supply and drainage system, and the monitoring system. The structure of the physical model is shown in Figure 3c.

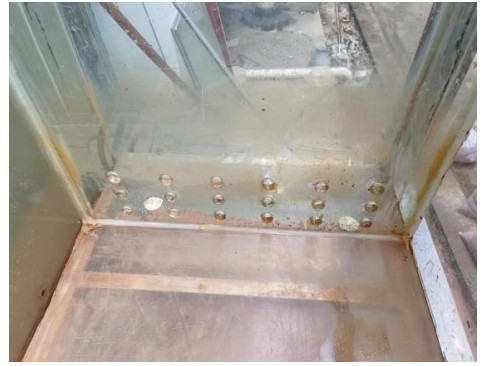
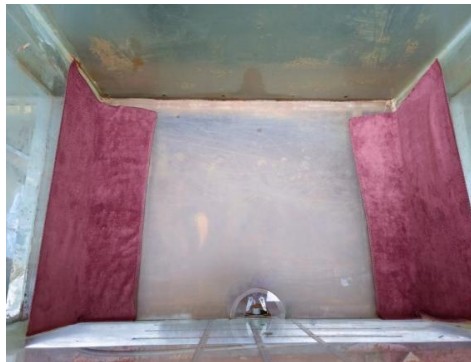

(**a**) Lateral weep holes　　　　　　　　(**b**) Prevention of loss of cover soil

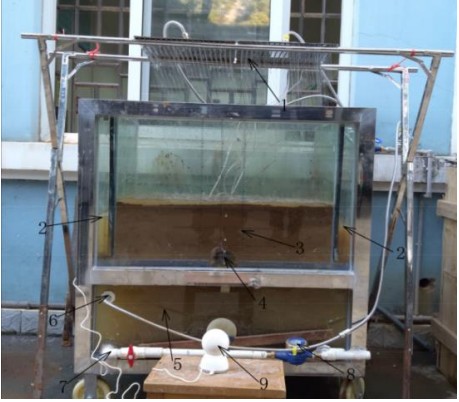

(**c**) Overall model drawing

**Figure 3.** Physical model installations. 1—Special rainfall shower 2—model unit side water tank. 3—overburden soil 4—existing earth cave. 5—dissolving chamber water tank 6—dissolution chamber inlet. 7—dissolution chamber drain 8—drainage monitoring water meters. 9—cameras.

### 3.2.1. Body of the Model

The main model box has dimensions of L (120 cm) × W (70 cm) × H (100 cm) and is made up of two parts, the upper and lower, as shown in Figure 3. The upper part of this consists of a central soil box (100 cm × 70 cm × 60 cm)) and water tanks (10 cm × 70 cm × 60 cm long) on the left and right sides. The middle soil box is used to simulate the Quaternary soil layer and the water tanks on either side are used to simulate other recharge sources. The lower part is a rectangular (120 cm × 70 cm × 40 cm) water tank, which is used to simulate the changing water level in the solution chamber. The toughened glass at the bottom of the cover soil is set with a semi-circular hole of 3 cm radius to simulate a soil cavity for the karst channel. The karst passages allow the water inside the cavity to communicate with the upper cover soil and thus interact and influence each other when the water level changes.

### 3.2.2. Rainfall Systems

The rainfall system in this experiment means it is difficult to control the intensity of rainfall accurately, and after reviewing the research results of relevant scholars, it was decided to use the total daily rainfall as the control factor for artificial rainfall [20,32]. The rainfall system is an independent external system, erected on a steel frame above the main model. A special showerhead is used to simulate artificial rainfall, with a switch, timer and water meter at the inlet pipe of the showerhead. The switch and timer control the timing of the rainfall and the water meter controls the daily rainfall amount.

### 3.2.3. Water Supply and Drainage Systems

The water supply and drainage system are divided into two parts, upper and lower, whose two systems are independent and do not affect each other. The upper left and right

tanks are designed to recharge the overburdened soil from the lateral direction, while the lower karst cavity water supply and drainage system are designed to provide water levels for the karst channels. The lower tank is connected to a water pipe connected to the mains as a water supply outlet pipe for the karst cavity and the rate and time of the water supply are controlled by a control switch and water meter. The lower tank is also equipped with a 100 mm diameter water pipe with a switch as a drainage pipe. The switch of the drain is set to close when the water is supplied to the karst cavity to simulate the rise of the water table in the karst area. When the drainage pipe is switched on for drainage, it simulates the fall of the water table (pumping or groundwater runoff) in the karst area and the amount of drainage can be measured by setting a flow meter. Throughout the test, the water supply and drainage can be regulated according to your test requirements.

### 3.2.4. Monitoring Systems

The deformation of the cover soil and the pressure of the karst cavity are monitored according to the test requirements of this paper. The displacement meter sensor model DMWY-100 is used for the determination of the cover soil deformation, with a compact size suitable for small-scale indoor model tests. The output is highly sensitive and can resolve displacement changes of less than 0.005 mm; has wide range, high accuracy, small drift, and stable performance; stainless steel full bridge housing, which is corrosion resistant and pressure resistant; technical parameters are shown in Table 1. The pressure in the karst cavity is monitored using a levelogger 5 barometric pressure sensor to monitor changes in pressure during supply and drainage. A video camera is used to record the deformation to collapse of the overburden on the karst bedrock, with a focus on capturing the deformation, intensification, and end of the karst collapse caused by rainfall and lateral recharge, water supply, and drainage.

**Table 1.** Technical parameters of DMWY-100 surface type displacement meter.

| Dimensions | $\Phi 25 \times 345$ | Measuring Range (mm) | 0~100 |
|---|---|---|---|
| Full output ($\mu\varepsilon$) | 8000 | Correction factor (mm/$\mu\varepsilon$) | 0.0125 |
| Precision | $\leq 0.2\%$F•S | Error (mm) | $\leq 0.01$ |

Note: F•S is the full-scale output value, around 8000 microstrain.

### 3.3. Test Program

The experiments in this paper focus on the effect of dynamic changes in groundwater levels on the deformation to collapse of overlying karst caps. Therefore, two different hydrodynamic models were developed: (1) the effect of different recharge conditions on the deformation to collapse of the cover soil; (2) the effect of groundwater fluctuation on the deformation to collapse of the cover soil. Physical model tests were also carried out for five different cover thicknesses under two hydrodynamic models, i.e., to quantitatively study the process of deformation to collapse of overlying karst cover. Their indoor physical model test protocols are shown in Table 2.

**Table 2.** Physical simulation test program for karst collapse.

| Test Program | Control Variables | Karst Opening Diameter/cm | Stratigraphic Conditions | Dry Density/ (g/cm³) | Initial Moisture Content/% |
|---|---|---|---|---|---|
| 1 | Rainfall and other recharge | 6 | Red clay Five different thicknesses | 1.40 | 30 |
| 2 | Dissolution chamber supply and drainage | 6 | Red clay Five different thicknesses | 1.40 | 30 |

### 3.4. Model Filling

Firstly, based on the moisture content $w_0 = 5\%$ measured after drying the soil samples taken from the study area, combined with the initial moisture content $w_1 = 30\%$ and dry density $p_d = 1.4$ g/cm$^3$ determined in the test protocol, and the different cover thicknesses required for each test (6 cm, 9 cm, 12 cm, 15 cm, 18 cm), the mass of soil and the mass of water required for each test were calculated respectively. The calculated mass of soil and the mass of water are then configured to obtain the soil required for the test. Finally, the configured soil samples are filled in batches in the model box. As both scenarios 1 and 2 were the same soil structure in 5 different thicknesses, the filling protocol was the same. For the first 6 cm soil thickness test, the soil sample was divided into two parts and filled in two batches, with one sample filled to 3 cm and the remaining sample filled to 6 cm with surface trimming. Each subsequent test was filled to the required height for each test, following the same filling method as the first 6 cm.

### 3.5. Different Recharge and Supply and Drainage Controls

3.5.1. Different Supplies

To study the evolution of the cover soil disaster under rainfall conditions, a total rainfall of 255.4 mm was selected as the basis for the artificial rainfall conditions. In order to study the deformation of the parameters during the rainfall cessation period, a 52 min rainfall cessation period was used during the rainfall process. The rainfall starts at 13:30 and is suspended from 17:24 to 18:16. The rainfall stops at 19:30 when the cumulative rainfall reaches 255.4 mm. Throughout the rainfall process, there is a period when rainfall is stopped in order to analyze the effect of rainfall on the trend of cover displacement and deformation. Figure 4 shows the cumulative rainfall time curve taken during one rainfall test.

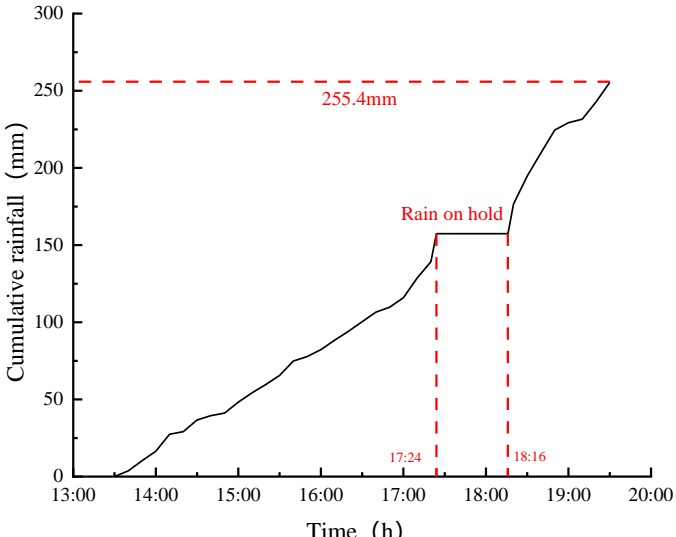

**Figure 4.** Rainfall vs. time curve.

3.5.2. Dissolution Chamber Supply and Drainage

The water level in the lower karst cavity was 4 cm before the model was filled. After the model was filled, the rainfall device was turned on to allow the covered soil to be fully saturated with water. After the model's seepage field had stabilized, the water level in the karst cavity was slowly raised, i.e., water was supplied at a rate of 1.2 cm/min. When the water supply level is flush with the bottom of the cover layer, the water supply is stopped and the deformation is left to stabilize for 15 min. The water level in the karst cavity is then slowly lowered, i.e., drained at a rate of 1.5 cm/min. When the drainage level reaches 4 cm at the bottom of the cavity, the drainage is stopped and the deformation is stabilized for 15 min. This was repeatedly cycled and the displacement and positive and negative

pressure in the cavity were monitored during the deformation to collapse process. A video camera was used to capture any changes or leakage in the karst channel throughout the supply and drainage test.

## 4. Analysis of Model Test Results

### 4.1. Effect of Different Recharge on Cover Deformation

The tests show that there is no obvious damage and deformation of the cover soil during the initial rainfall and lateral recharge, only a significant rise in the water level of the cover soil, but scattered soil in the karst channels while the rainfall and lateral recharge infiltrate the soil. With the increase of cumulative recharge and continuous infiltration, the soil above the karst channels gradually forms soil cavities in the process of continuous descent. After the formation of the soil cavities, the continuous infiltration of recharge continues to raise the water level of the cover soil. The infiltration of soil water from the karst channel into the underlying karst cavity intensifies, and when there is a sudden change in the infiltration flow, the infiltrated water carries a small amount of soil particles, but when the flow is stable, the water is clearer. However, after the formation of the soil cavity, the continuous rainfall accelerates the destructive deformation of the soil and accelerates the upward development of the cavity. Taking the first recharge at the beginning of the test as an example, the relationship between the magnitude of displacement of each cover layer by extracting displacement meter data at the end of the test is shown in Figure 5: 6 cm (1.44 mm) > 9 cm (1.13 mm) > 12 cm (0.69 mm) > 15 cm (0.58 mm) > 18 cm (0.45 mm), the data indicates that for this cover type karst collapse, the thicker the cover layer the smaller the deformation displacement of the cover soil. However, the subsequent cumulative recharge will cause the cumulative deformation of the cover to become more and more severe until the collapse eventually occurs.

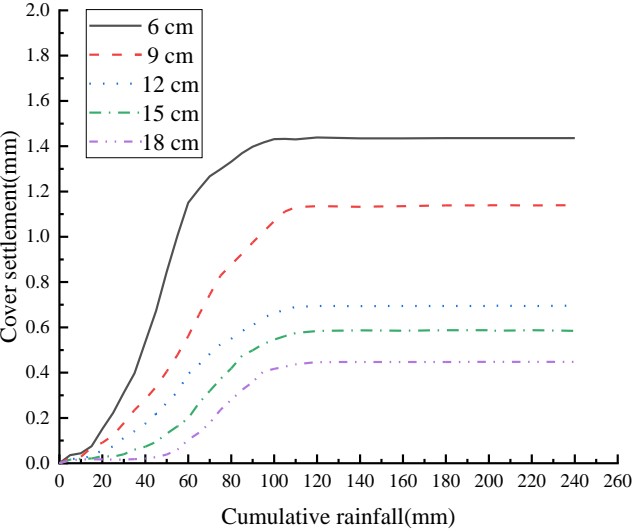

**Figure 5.** The relationship between rainfall and caprock deformation.

### 4.2. Effect of Groundwater Level Fluctuations on Cover Deformation

4.2.1. Analysis of the Influence of Water Level Rise on the Dissolution Cavity

The case in which the caprock density is $1.4 \text{ g/cm}^3$ and the average water supply rate is 1.2 cm/min was simulated. That is, when the positive pressure change in the dissolution chamber is the same, for different thicknesses of 6 cm, 9 cm, 12 cm, 15 cm, and 18 cm, the impact test of the karst collapse of the caprock in the karst cavity is shown in Figure 6.

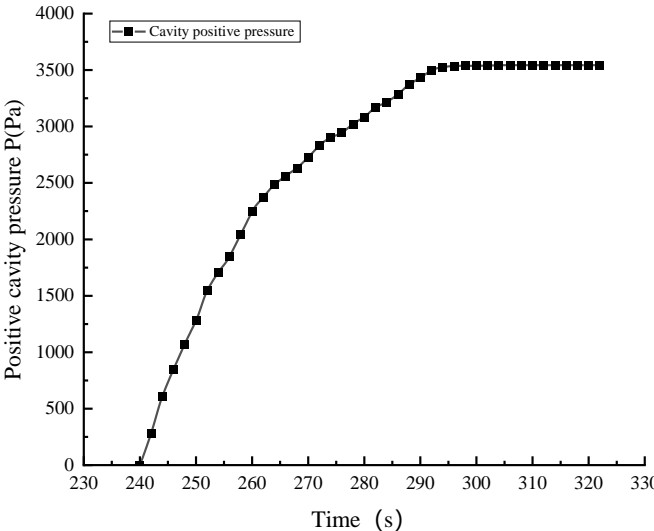

**Figure 6.** Time-lapse curve of the water supply and positive pressure of the dissolution chamber.

From Figure 6, it can be observed that there was no vacuum negative pressure in the dissolution chamber before 240 s. This is because the water inlet valve was closed and the pressure in the dissolution chamber was consistent with the external atmospheric pressure during this period. After 240 s, the water inlet valve was opened, and the instrument began to supply water, resulting in the positive pressure in the chamber. When the water supply continued, the groundwater level continued to rise, the space in the chamber continued to decrease, and the positive pressure value continued to increase. After 295 s, the positive pressure of the dissolution chamber reached the maximum of 3542 Pa. When the instrument started to supply water, the positive pressure of the dissolution chamber increased at the fastest rate of 375 Pa/s. With the increase of time, the positive pressure change rate gradually decreased. When the time was greater than 295 s, the positive pressure in the dissolution chamber tended to be stable, and the change rate was 0. This is mainly because with an increase in positive pressure in the cavity, the positive pressure acting on the cap layer gradually increases. When the pressure increases to a certain extent, the particles of the cap layer are connected to the outside pores, so that the gas in the cavity can be discharged. When the volume of water supplied is equal to that of the gas discharged from the cavity, the pressure in the cavity remains unchanged.

Figure 7 shows the deformation time curve of the caprock. It can be seen that the deformation-time curve of the caprocks with different thicknesses is similar to the positive pressure curve in the cavity, whereas the time required for the stability of caprock deformation is greater than that of the positive pressure curve. That is, when the positive pressure reaches the maximum, it still affects the deformation of the cover layer soil. After a period, the stress state of the cover layer reaches equilibrium again, and the cover layer soil no longer deforms. Before starting the water supply, the covering layer does not deform, the deformation amount is almost 0, and the structure is relatively stable. When the water supply starts, the covering layer begins to deform slowly. In addition, with a continuous water supply, the amount of deformation gradually increases, the final deformation is unchanged, and the structure tends to be stable.

When the thickness of the covering layer is 6 cm, the covering layer is stable at 376 s, and its deformation is 3.016 mm. When the covering layer thickness is 18 cm, the deformation of the covering layer remains unchanged at 342 s, and its deformation is 0.526 mm. When the caprock density and positive pressure in the cavity are the same, with an increase in caprock thickness, the caprock deformation decreases continuously. Namely, the caprock thickness is inversely proportional to the caprock deformation (Figure 8). In the meantime, when the thickness of the cap layer is larger, the deformation rate in the deformation stage is smaller. When the thickness of the cap layer increases from 6 cm to

18 cm, the deformation rate decreases from 0.132 mm/min to 0.0283 mm/min. This result indicates that the larger the cap layer thickness is, the more difficult it is for the cap layer to deform and fail, and the more stable the structure is.

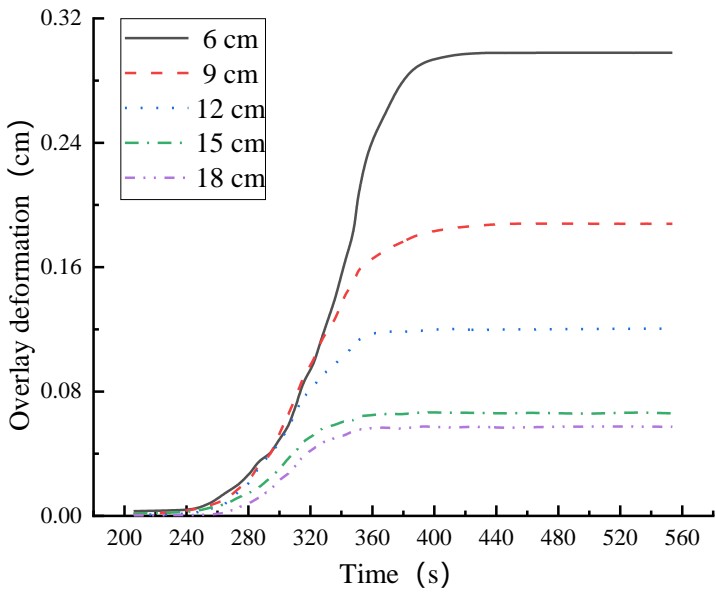

**Figure 7.** Rising water table cover settlement.

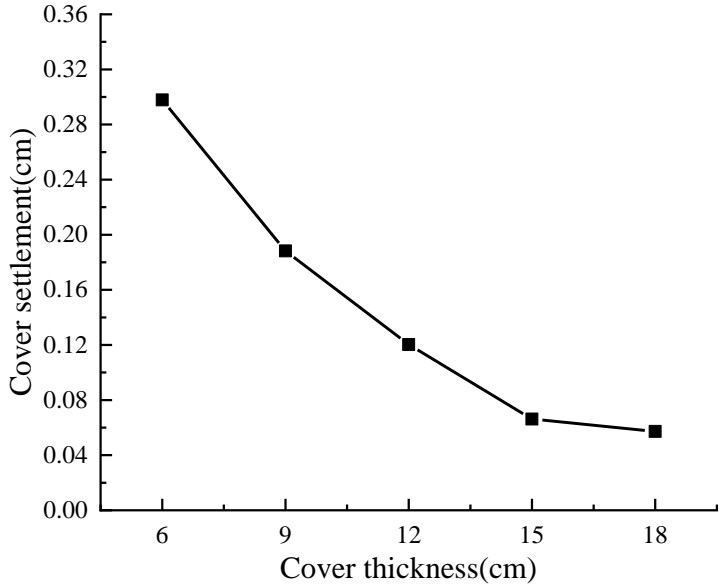

**Figure 8.** Settlement curve for each cover layer during water level rise.

4.2.2. Analysis of the Results of the Dissolved Cavity Water Level Drop Test

In the simulation test, the density of the cap layer is 1.4 g/cm$^3$ and the drainage rate is 1.5 cm/min. That is, with the same change of vacuum negative pressure in the dissolution chamber, different thicknesses of 6 cm, 9 cm, 12 cm, 15 cm, and 18 cm are simulated via experiments to study the effect of caprock on karst collapse. The variations of air pressure in the cavity are shown in Figure 9.

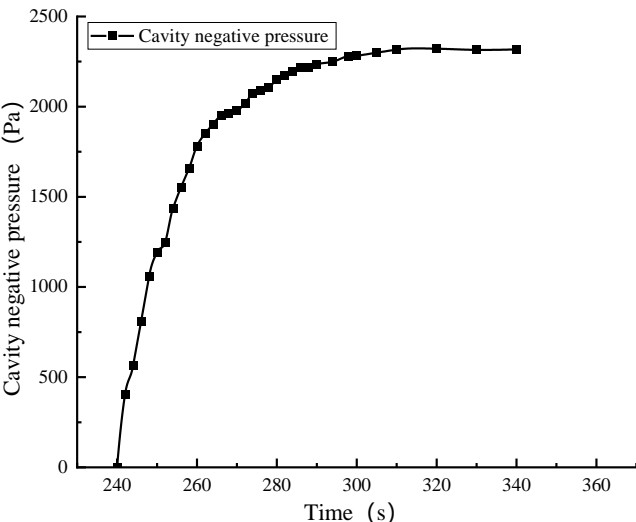

**Figure 9.** Time-lapse curve of the drainage and cavity pressure.

It can be seen from Figure 9 that before 240 s, there is no vacuum negative pressure in the dissolution chamber in which the pressure is equal to the external atmospheric pressure. At 240 s, the dissolution chamber drainage switch is turned on to start drainage. The continuous process makes the space in the dissolution chamber increase continuously, and the vacuum negative pressure also keeps increasing. When the test goes on for 310 s, the vacuum negative pressure reaches the maximum of 2321 Pa. After drainage, the vacuum negative pressure increases rapidly during 240–270 s. After 270 s, the vacuum negative pressure increases slowly with the increase of time. When the time is greater than 310 s, the negative pressure in the dissolution chamber tends to be stable. Due to the pressure difference between the top and bottom of the cap layer, the vacuum suction force acting on the cap layer gradually increases with the vacuum negative pressure. The external air enters the karst cavity through the connected pores. When the volume of the discharged water is equal to that of the gas entering from the outside, the negative pressure in the karst cavity remains constant.

Figure 10 shows the deformation-time curve of the caprock. Before drainage, the cover layer soil is relatively stable and hardly deforms, and the deformation amount is approximately 0; after the drainage starts, the cover layer soil begins to deform slightly. With the increase in drainage time, the cumulative deformation of the covered soil also increases. As the test continues, the cover soil no longer deforms when other conditions remain unchanged, and finally tends to be stable. When the thickness of the cover layer is 6 cm, the deformation rate in the curve deformation stage is the largest. As the thickness of the cover layer increases, the deformation rate of each curve becomes smaller and smaller, indicating that the deformation and displacement of the cover layer are inversely proportional to its thickness. That is, the thicker the cover layer, the more difficult it is to deform and break. It can be seen from Figure 11 that under the same negative pressure of the cavity, the larger the thickness of the overburden, the smaller the deformation, and the more stable the overburden soil is. the shear strength increases accordingly, and the ability of the cover layer to resist deformation and damage also increases. The deformations of the cover may have a nonlinear relationship with its thickness. With the increase of cover thickness, the change rate of soil deformation gradually slows down.

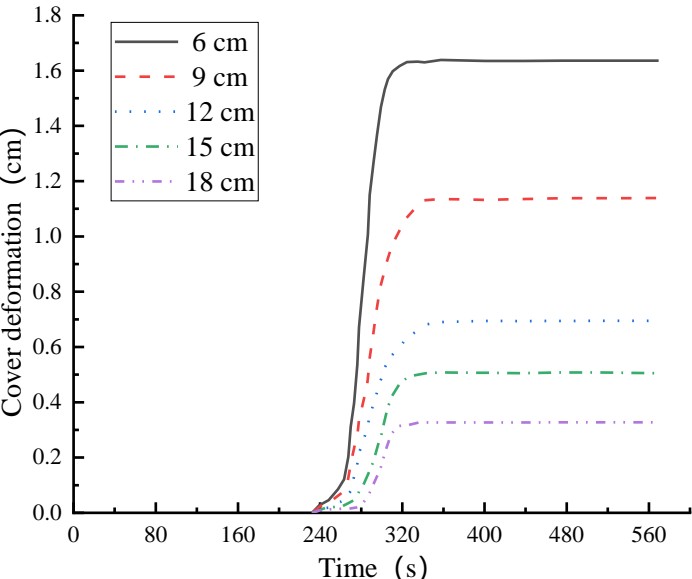

**Figure 10.** Declining water table cover settlement.

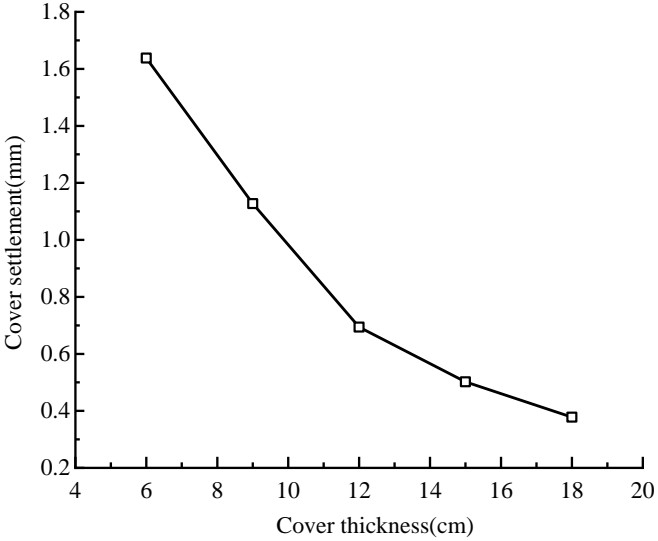

**Figure 11.** Settlement curves for each cover layer as the water level drops.

### 4.3. Analysis of the Collapse Development Process

The surface water, pore water, and groundwater in the lower part of the soil box are closely linked to form a complete hydrological cycle in the test model installation. The infiltration of surface water can change the water content of the cover soil. When rainfall or other recharge infiltrates, the water content of the cover soil increases, the cohesion and angle of internal friction decrease, the capacity and self-gravity increase, and the shear strength decreases, which ultimately leads to a reduction in collapse resistance. The effect of groundwater on karst ground collapse can be analyzed in many ways, but the magnitude and frequency of groundwater fluctuations are the most important. Based on the development of the soil cavity under the two test scenarios the whole process can be roughly divided into a softening phase of the soil above the cavity, a soil loss-spalling phase and a rapid-integral collapse phase (Figure 12).

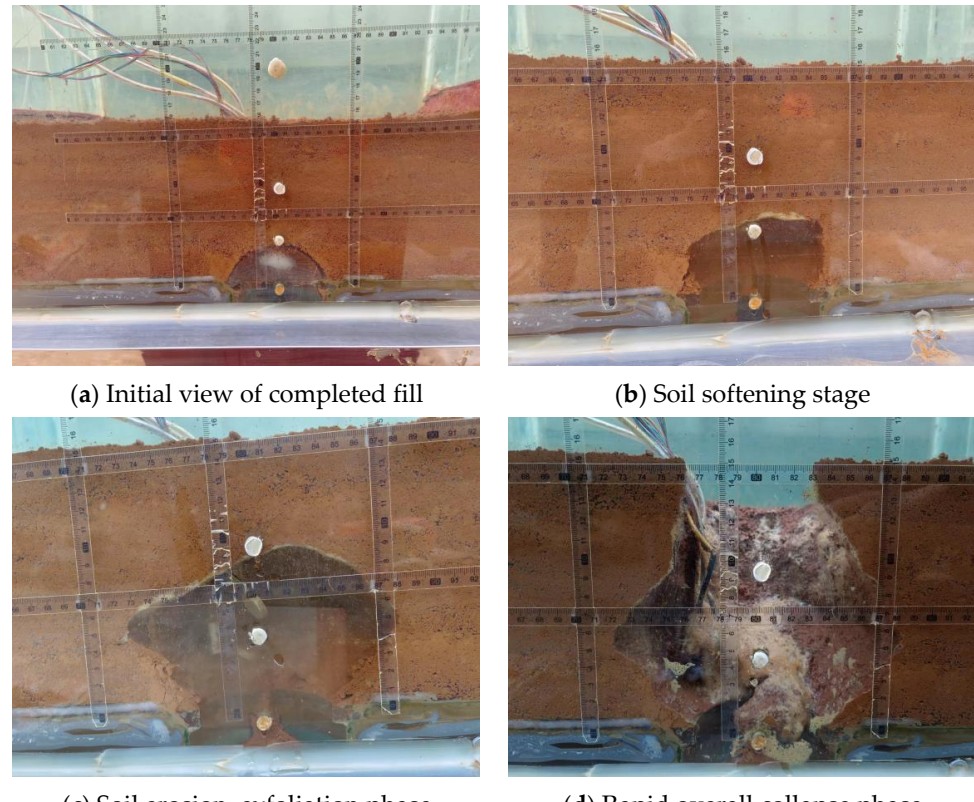

(**a**) Initial view of completed fill  (**b**) Soil softening stage

(**c**) Soil erosion–exfoliation phase  (**d**) Rapid overall collapse phase

**Figure 12.** Schematic diagram of the development of the earth cave collapse.

(1)  Soil softening stage: At the beginning of the test, rainfall and lateral seepage recharge the groundwater and a gradient difference is formed between pore water and groundwater, resulting in vertical seepage. As the support for the cover soil above the soil hole is weak, it is susceptible to submerged erosion making the soil particles soften.

(2)  Soil loss exfoliation stage: With the increase of rainfall and lateral seepage recharge time, the saturation of the soil increases and the permeability becomes poor. When the water level rises, the gas inside the cavern cannot be discharged and is squeezed to form a high-pressure air mass, which produces tiny fissures in the cover soil. When the water level falls, the negative pressure inside the cavern increases the additional force of downward seepage of pore water in the cover soil, which intensifies the loss and spalling of the soil.

(3)  Rapid integral collapse phase: long periods of continuous recharge and fluctuations in the water table resulted in a large loss and spalling of the cover soil above the cavity in the later stages of the test, and the arch structure no longer existed. The cavity was observed earlier in the form of an altar shown in Figure 13, at which point the self-weight of the soil increased with recharge, while the thickness of the soil was still being lost and spalled with fluctuations in the water table. When the collapse develops to the point where the collapse causing force is greater than the collapse resisting force, the soil undergoes shear deformation falling off as a whole, eventually leading to the overall collapse of the surface, as shown in Figure 14.

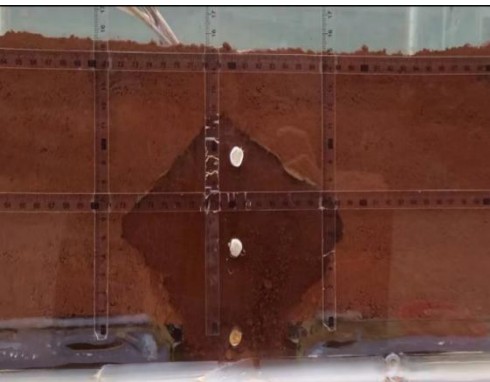

**Figure 13.** Cavity altarpiece.

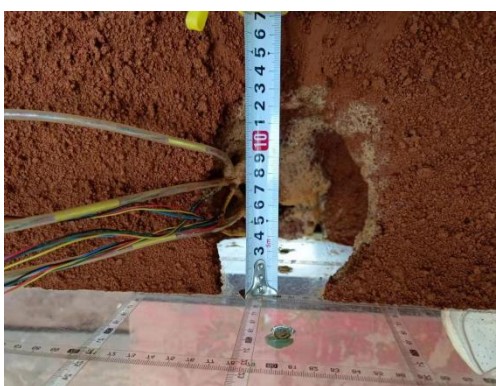

**Figure 14.** Surface collapse map.

## 5. Conclusions

In this paper, based on the results of previous studies on overlying karst in Guilin, indoor model tests on karst cover deformation under groundwater level fluctuations were carried out, which mainly led to the following conclusions.

(1) Under the action of rainfall and other recharge, cavity supply, and drainage, the infiltration curve of the soil in the cover layer changes faster at the beginning and slows down gradually at the later stage as the time and intensity of rainfall and other recharge increase. The thicker the cover layer, the slower the overall deformation; when the rate of cavity supply and drainage is certain, the thicker the cover layer, the smaller the deformation produced by the fluctuation of the groundwater level; after the formation of the cavity, rainfall, and other recharge, cavity supply and drainage accelerate the deformation of the soil and the upward development of the cavity.

(2) Under the condition of uniform initial water content and density, the cumulative amount and time of rainfall and other recharge required for the deformation to collapse process increases as the thickness of the cover soil increases, and the number of cycles of cavity drainage required also increases. This indicates that the fluctuation frequency of groundwater level can aggravate the expansion of karst collapse.

(3) The karst cover is affected by a combination of self-gravity, subduction by surface water seepage, suction by water-air change and airburst during the deformation to collapse process. For the same thickness of cover soil, the relationship between the deformation due to different effects is: water level rise is greater than water level fall which is greater than rainfall and lateral recharge; for example, with a cover of 6 cm, water level rise (3.04 mm) > water level fall (1.63 mm) > rainfall and lateral recharge (1.44 mm).

**Author Contributions:** Conceptualization, X.C., Y.S. and X.G. (Xiaotong Gao); methodology, X.G. (Xiaotong Gao); validation, H.L., X.G. (Xiaohui Gan) and M.X.; investigation, X.G. (Xiaotong Gao); writing—original draft preparation, X.G. (Xiaotong Gao); writing—review and editing, Y.S.; supervision, H.L. and X.G. (Xiaohui Gan); project administration, X.C. All authors have read and agreed to the published version of the manuscript.

**Funding:** (1) National key research and development program subject "R&D and experimental demonstration of key technologies for water resource regulation in Karst wetlands in the Lijiang River Basin" (2019YFC0507502). (2) The National Natural Science Foundation of China Project "Research on the Collapse Mechanism of Karst Water-soil Coupling in Guilin under Extreme Climate Conditions" (41967037). (3) Guangxi Science and Technology Program Project "Wetland Water Resources Utilization and Water Ecology" (RZ2100000161).

**Institutional Review Board Statement:** Not applicable.

**Informed Consent Statement:** Not applicable.

**Data Availability Statement:** Not applicable.

**Conflicts of Interest:** The authors declare no conflict of interest.

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
