# Peer review of "Model Test Analysis of Groundwater Level Fluctuations on Karst Cover Deformation Taking the Monolithic Structure of Guilin as an Example"

_applsci, doi:10.3390/app13031747_

Round 1

Reviewer 1 Report

Main remarks:

The article concerns the issue of the influence of water level fluctuations in the karst cover on its deformation. Model tests were carried out in a specially built apparatus in which the deformation of the ground layer subjected to rainwater infiltration was simulated. The most significant element of the study was a 3 cm hole that simulated a soil cavity and the karts channel. The deformation of the ground layer of various thicknesses (6 - 18 cm) under constant rainfall was analyzed. In order to explain the phenomenon, plots of water pressure changes in the pores versus time with the effect of rainfall were also presented. The presented plots, as well as other test results, have been analyzed in detail. The presented research results can be treated as a preliminary diagnosis of the problem, which in the future may constitute important experimental data for modeling the describing phenomenon.

Critical remarks:

The article concerns strictly model tests (laboratory, performed on a small-scale model), hence the content of the article does not contain any statements suggesting that a model of the phenomenon is being analyzed. For example, the title of chapter 2.1 is incorrect - Fig. 2 does not show the physical model, but the apparatus for model testing. Similarly, line 363 (conclusions) does not generalize the geological model.

Often in the article the word "deformation" appears incorrectly. As far as the deformation of the layer is mentioned (e.g. the method of deformation of the soil layer), this expression is appropriate, but if there are values given in units of length (cm or mm), then the word “displacement” or “settlement” should be used instead of the word "deformation".

The presented references also raises doubts. While all items are in English, their authors are from China. In order for an article to be published in a worldwide journal, the bibliography should also include items presenting the state of knowledge in other countries.

Detailed remarks:

– Figure 2: incomprehensible relation between cover deformation and cumulative rainfall; the presented plot looks like a cover deformation vs time relationship; however, if the authors really wanted to show the relationship between cover deformation and cumulative rainfall, it should be noted that the relationship depends very much on the rate of rainfall and in the general case the plot may have a significantly different shape,

– Lines 376-377: repeating part of the previous sentence.

Author Response

 In response to your questions about the title and the lack of geological model generalisation, I have revised and added geological model generalisation.

3.1.  Generalisation of the geological model of the study area

Due to the complex geological conditions in the study area, there are many uncertainties. If the karst-induced collapse test were to be carried out directly at the study site, it would be constrained by many factors. It is therefore difficult to carry out in-situ karst collapse tests at the site. In order to obtain the results of the karst collapse test under changing groundwater levels and to carry out the test successfully, a geological model of the geological conditions in the study area needs to be generalised here.

The geological drilling data collected from Lingui District, Guilin City, shows that the overlying karst in Lingui District can be mainly classified into single-layer structure or double-layer structure. Among the monolayer structures, the overlying soil layer is clayey. In this simulation test study, a specific study area was selected to focus on the monolithic structure (Figure 2).

Reviewer 2 Report

The manuscripts present experimental results analyzing different factors of overburden collapse above karst cavity. Different conditions, mainly rainfall and groundwater level fluctuations, are investigated.

I find this manuscript interesting and believe there is scientific contribution, but additional modifications and clarifications are needed before recommending for publishing. Thus, I recommend major revision.

Main comments:

1.     I must notice that almost all references are confined to work from China. I would find it extremely surprising if there is no any related work done abroad. Thus, I believe that more extensive literature overview and clear relation with other work in the field is needed (particularly on existing karst physical models related to the topic).

2.     Is the karst opening from Table 1. the same as semi-circular hole defined at line 117? Please clarify better these two in your manuscript.

3.     Correct title of section 3.2. (it is the same as 3.1.).

4.     Define position of the pressure sensor in the cavity. Denote point when chamber is filled with water on Figure 4. (i.e. when water reaches cover layer). Is it around 265 sec?

5.     Experiment 3.2.1. is unclear to me. I assume that outfall/drainage pipe is closed. There is water inflow in initially air-filled cavity and water tanks in upper part of model are empty? Is water inflow stopped at some point or is it that water overflows to water tanks at some point (when there is no increase in pressure anymore)? Better explanation and more details are needed.

6.     Please explain experiment 3.2.2. in more details as well.

7.     I do not see any relation between title of 3.2.2. section and its content. The section 3.2.2. investigates effect of cavity underpressure on cover deformation for different cover thicknesses. Even that contaminant concentration and soil compressive strength could be indirectly related, they are nowhere mentioned or discussed in text. Change title appropriately.

 Minor comments:

1.     Ln. 24: “cover layer is not as thick as the cover layer.” Make this sentence clearer.

2.     Ln 128. Add reference.

3.     More detailed image or better (resolution) picture (1b) of physical model would be nice. Also specifying dimensions and more details on 1a.

4.     Please define differences between outflow/drainage and overburden/cover/solum/ caprock. Avoid switching different words for the same thing if that is the case.

5.     More details regarding used red clay is needed.

6.     As authors did use video camera during experiment, I believe it would be very beneficial to include some images to improve overall presentation of the manuscript. (particularly pictures during collapse).  

7.     Description of physical model could include more details (and images).

8.     Analysis of “other recharge” is actually missing. Section 3.1.2 doesn’t produce any meaningful results. I would recommend either reproduce cyclic study or not mentioning it throughout whole manuscript.

9.     The terms positive and negative pressures are stated multiple times in manuscript. I believe this is relative pressure against atmospheric pressure. Please state this more carefully, particularly as gas (absolute) pressure cannot be negative. Reconsider using other terms, for example overpressure and underpressure instead, if you find them more appropriate.

10.  Consider adding word “physical” or “experimental” model/modeling in your title to make it clearer.   

Reviewer 3 Report

This is an interesting paper on a subject that should be of great interest to many readers. However, it is not yet suitable for publication. Significant revisions are required focusing the scope of the study, thoroughness and scientific discussions. For location of comments, see the attached file.

Author Response

Point 2: Information about the method is too short and poorly explained, i.e., why the authors used five different cover thicknesses, the same density, and other assumptions. The result needs more quantitative information (with exact values based on the model, uncertainty, etc.) rather than qualitative information.

Response 2: A description of the selected parameters has been added to the text, and the conclusions in the abstract have been recapitulated and revised.

Reviewer 4 Report

I read the manuscript submitted by Chen et al. for consideration in Applied Sciences with great interest. This manuscript presents an idea regarding the model test analysis of groundwater level fluctuations on karst cover deformation. Although the manuscript presents a good dataset and addresses relevant research questions, it cannot be accepted for publication in its present form.

As detailed in my comments hereafter, the manuscript needs a profound revision in form. More fundamentally, the manuscript has no real discussion yet and also, where interpretations are expected, lacks detailed referencing to demonstrate that the author tries to put their results in context. It is sad as the author's work could be much better presented, supported, and discussed contribution. The manuscript needs to have a proper, scientifically correct discussion and more in-depth arguments for the style of the model test analysis of groundwater level fluctuations on karst cover deformation, including model validation, accuracy, and uncertainty (if any).

Title

Is this model for karst in general or in a specific area? Please add that information to the title if it is based on a particular site.

Abstract

Information about the method is too short and poorly explained, i.e., why the authors used five different cover thicknesses, the same density, and other assumptions. The result needs more quantitative information (with exact values based on the model, uncertainty, etc.) rather than qualitative information.

Introduction

-       Several specific statements need references to support the arguments, i.e., lines 37-40; lines 40-44; 64-65; 75-77;

-       Lines 48-95: the paragraph is too thick and may be hardly followed by the readers. Please divided into 3-4 paragraphs.

-       Lines 77-89: Please summarize it, compare each other, and only show the most exciting and essential findings rather than explain their research one by one in a long paragraph. Based on the summary, the authors should put their research in a specific position where they will contribute to the topics.

-       L96-103: The authors should end their introduction with a strong, straightforward, and clear statement. The final paragraph of their introduction should elaborate in more detail, forecast their main arguments and conclusions, and provide a brief description of the rest of the paper that lets the reader know where they are going and what to expect. In addition, concluding the introduction with an explicit roadmap tells the reader that the authors clearly understand their paper's structural purpose.

Methods

-          Lines 109-119: There is no explanation for why the authors used those specifics and dimensions for the model's body. Furthermore, Figure 1 is poorly illustrated (especially 1b) and poorly explained in the text (1a and 1b).

-          Lines 127-128: Which scholars and what considerations that the authors take?

-          Lines 134-146: There is no explanation for why the authors used those specifics and dimensions for the model's body, as well as which references the authors used to build the water supply and drainage systems.

-          Lines 149-151: Please elaborate on why the authors used these tools.

-          Lines 160-162: Please elaborate on why the authors used five different cover thicknesses under two hydrodynamics models; which references that they use?

-          Table 1: Please elaborate on each column's consideration selection of data or parameters.

-          Why are the materials red clay soils? Please elaborate in more detail. Also, if possible, the authors can show using a map where they took the sample for their simulation.

-          Since this is a model, the authors should explain in detail why they used those specific parameters and/or dimensions and which references they used to build that model. So far, we cannot see that information in the methods section. Furthermore, the most important things, if we talk about the model, are uncertainty and error. The authors did not give this information on how they validate their model and how they know whether the model is accurate or if there is uncertainty.

-          If possible, please add a chart/diagram to summarize your (revised) methods from the start (data, input, etc.) to the end/output, including the (possible) problem that may exist during the simulation.

Results (and discussions)

The results part is too long, and somehow it is just a repetition of the illustration in the figures/table. It will be better if the authors can make it simple and efficient with not too much text. Also, it would be better if they could separate the Results and Discussion parts. In this version, the manuscript has no real discussion yet, and also, where interpretations are expected, it lacks detailed referencing to demonstrate that the author tries to put their results in context. It is sad as the author's work could be much better presented, supported, and discussed contribution. The manuscript needs to have a proper, scientifically correct discussion and more in-depth arguments for the style of the model test analysis of groundwater level fluctuations on karst cover deformation, including model validation, accuracy, and uncertainty (if any).

Author Response

Response to Reviewer 4 Comments

I read the manuscript submitted by Chen et al. for consideration in Applied Sciences with great interest. This manuscript presents an idea regarding the model test analysis of groundwater level fluctuations on karst cover deformation. Although the manuscript presents a good dataset and addresses relevant research questions, it cannot be accepted for publication in its present form.

As detailed in my comments hereafter, the manuscript needs a profound revision in form. More fundamentally, the manuscript has no real discussion yet and also, where interpretations are expected, lacks detailed referencing to demonstrate that the author tries to put their results in context. It is sad as the author's work could be much better presented, supported, and discussed contribution. The manuscript needs to have a proper, scientifically correct discussion and more in-depth arguments for the style of the model test analysis of groundwater level fluctuations on karst cover deformation, including model validation, accuracy, and uncertainty (if any).

Point 1: Is this model for karst in general or in a specific area? Please add that information to the title if it is based on a particular site.

Response 1:  Amend the title as follows:

Model test analysis of groundwater level fluctuations on karst cover deformation

——Take the monolithic structure of Guilin as an example

Point 2: Information about the method is too short and poorly explained, i.e., why the authors used five different cover thicknesses, the same density, and other assumptions. The result needs more quantitative information (with exact values based on the model, uncertainty, etc.) rather than qualitative information.

Response 2: A description of the selected parameters has been added to the text, and the conclusions in the abstract have been recapitulated and revised.

3.2. Physical modelling principles

When conducting physical model tests, not only should the test requirements and operability be met, but similar issues should also be considered. According to the geological reports available in the study area, it is known that the thickness of the collapse prone overburden in the study area ranges from 0.7 to 5.6m, and the diameter of the karst openings on the bedrock surface varies from 0.5 to 1.2m. The geometric similarity ratio was finally determined to be 1:15 by combining the geological, hydrological and tectonic conditions of the study area. 6 cm diameter of the reserved soil hole was set, and the thickness of the overburden layer was 6 cm, 9 cm, 12 cm, 15 cm and 18 cm. The soil samples of the overburden layer were taken directly from the site during the test, and the moisture content of the test was close to that of the site. However, the density was difficult to achieve in situ, and after repeated indoor tests, the density was finally determined to be 1.4 g/cm³. In order to make the simulation conditions as close to the site as possible, to take into account the factors involved in karst collapse at the site and to ensure that the soil simulation parameters are basically the same as the original soil at the site, the karst collapse test device was specially tailored. In the course of the test, the test device is subjected to a large load when filling a predetermined thickness of overlying soil. This is why the frame of the device is made of a high hardness aluminium alloy and the side walls are inlaid with 5mm thick toughened glass. In order to have a clear view of the deformation and damage process inside the cover soil, the karst access opening was set up in a semi-circle at the midpoint of the front side. A special ice block was pre-buried above the passage opening to form a pre-existing soil cavity in the shape of a 1/4 sphere during the test. The physical model consists of four parts: the main model, the rainfall system, the water supply and drainage system and the monitoring system. The structure of the physical model is shown in Figure 3.

Point 3: Introduction

-  Several specific statements need references to support the arguments, i.e., lines 37-40; lines 40-44; 64-65; 75-77;

-  Lines 48-95: the paragraph is too thick and may be hardly followed by the readers. Please divided into 3-4 paragraphs.

-  Lines 77-89: Please summarize it, compare each other, and only show the most exciting and essential findings rather than explain their research one by one in a long paragraph. Based on the summary, the authors should put their research in a specific position where they will contribute to the topics.

-  L96-103: The authors should end their introduction with a strong, straightforward, and clear statement. The final paragraph of their introduction should elaborate in more detail, forecast their main arguments and conclusions, and provide a brief description of the rest of the paper that lets the reader know where they are going and what to expect. In addition, concluding the introduction with an explicit roadmap tells the reader that the authors clearly understand their paper's structural purpose.

Response 3: I have completed the changes in the article in response to the problems you pointed out in the introduction.

Point 4: Methods

-  Lines 109-119: There is no explanation for why the authors used those specifics and dimensions for the model's body. Furthermore, Figure 1 is poorly illustrated (especially 1b) and poorly explained in the text (1a and 1b).

-  Lines 127-128: Which scholars and what considerations that the authors take?

-  Lines 134-146: There is no explanation for why the authors used those specifics and dimensions for the model's body, as well as which references the authors used to build the water supply and drainage systems.

-  Lines 149-151: Please elaborate on why the authors used these tools.

-  Lines 160-162: Please elaborate on why the authors used five different cover thicknesses under two hydrodynamics models; which references that they use?

-  Table 1: Please elaborate on each column's consideration selection of data or parameters.

-  Why are the materials red clay soils? Please elaborate in more detail. Also, if possible, the authors can show using a map where they took the sample for their simulation.

-  Since this is a model, the authors should explain in detail why they used those specific parameters and/or dimensions and which references they used to build that model. So far, we cannot see that information in the methods section. Furthermore, the most important things, if we talk about the model, are uncertainty and error. The authors did not give this information on how they validate their model and how they know whether the model is accurate or if there is uncertainty.

-  If possible, please add a chart/diagram to summarize your (revised) methods from the start (data, input, etc.) to the end/output, including the (possible) problem that may exist during the simulation.

Response 4:Methods

 The main reason for these questions in the methods section is that in the previous article I did not explain the geological background of the study area and did not make a geological overview. Where the previous explanation of the subject model was inadequate, a more detailed explanation has been added, and a corresponding geological map of the study area has been added.

2.2.Geological features

The geological drilling data available in the study area shows that the Quaternary strata in the study area are widely distributed, mainly consisting of red clay formed by the Upper Pleistocene Alluvium (Q3al-pl), the Upper Pleistocene Residual Slope Formation (Q3dl-el) and the Upper Devonian Rongxian Formation (D3r). The Upper Pleistocene Alluvium (Q3al-pl) is a yellow and tawny pebble-gravel sandy clay and chalky clay, composed of gravel, clay and a small amount of fine sand and pebbles, 2-14 m thick; the Upper Pleistocene Residual Slope Formation (Q3dl-el) is a brick-red and light yellow clay, sandy clay and sandy-clay gravel layer, containing a small amount of quartz particles, 0.1-10 m thick; the Upper Devonian Rongxian Formation (D3r) is greyish-white, slightly weathered, cryptocrystalline in structure, medium-thick laminated. -The main mineral is calcite, with calcite veins developed and tightly cemented, the rock is relatively intact, the core is columnar, the bedrock surface within the site is highly undulating, and the karst fissures are relatively developed (Figure 1).

Figure 1.  Engineering Geological Map

  1. Test materials and methods

3.1.  Generalisation of the geological model of the study area

Due to the complex geological conditions in the study area, there are many uncertainties. If the karst-induced collapse test were to be carried out directly at the study site, it would be constrained by many factors. It is therefore difficult to carry out in-situ karst collapse tests at the site. In order to obtain the results of the karst collapse test under changing groundwater levels and to carry out the test successfully, a geological model of the geological conditions in the study area needs to be generalised here.

The geological drilling data collected from Lingui District, Guilin City, shows that the overlying karst in Lingui District can be mainly classified into single-layer structure or double-layer structure. Among the monolayer structures, the overlying soil layer is clayey. In this simulation test study, a specific study area was selected to focus on the monolithic structure (Figure 2).

Figure 2. Geological generalisation of a single structure

1—Overburden clayey soil;2—Limestone;3—Caves

3.2. Physical modelling principles

When conducting physical model tests, not only should the test requirements and operability be met, but similar issues should also be considered. According to the geological reports available in the study area, it is known that the thickness of the collapse prone overburden in the study area ranges from 0.7 to 5.6m, and the diameter of the karst openings on the bedrock surface varies from 0.5 to 1.2m. The geometric similarity ratio was finally determined to be 1:15 by combining the geological, hydrological and tectonic conditions of the study area. 6 cm diameter of the reserved soil hole was set, and the thickness of the overburden layer was 6 cm, 9 cm, 12 cm, 15 cm and 18 cm. The soil samples of the overburden layer were taken directly from the site during the test, and the moisture content of the test was close to that of the site. However, the density was difficult to achieve in situ, and after repeated indoor tests, the density was finally determined to be 1.4 g/cm³. In order to make the simulation conditions as close to the site as possible, to take into account the factors involved in karst collapse at the site and to ensure that the soil simulation parameters are basically the same as the original soil at the site, the karst collapse test device was specially tailored. In the course of the test, the test device is subjected to a large load when filling a predetermined thickness of overlying soil. This is why the frame of the device is made of a high hardness aluminium alloy and the side walls are inlaid with 5mm thick toughened glass. In order to have a clear view of the deformation and damage process inside the cover soil, the karst access opening was set up in a semi-circle at the midpoint of the front side. A special ice block was pre-buried above the passage opening to form a pre-existing soil cavity in the shape of a 1/4 sphere during the test. The physical model consists of four parts: the main model, the rainfall system, the water supply and drainage system and the monitoring system. The structure of the physical model is shown in Figure 3.

1—Special rainfall shower                         2—Model unit side water tank

3—Overburden soil                               4—Existing earth cave

5—Dissolving chamber water tank                  6—Dissolution chamber inlet

7—Dissolution chamber drain                      8—Drainage monitoring water meters

9—Cameras

Point 5: Results (and discussions)

The results part is too long, and somehow it is just a repetition of the illustration in the figures/table. It will be better if the authors can make it simple and efficient with not too much text. Also, it would be better if they could separate the Results and Discussion parts. In this version, the manuscript has no real discussion yet, and also, where interpretations are expected, it lacks detailed referencing to demonstrate that the author tries to put their results in context. It is sad as the author's work could be much better presented, supported, and discussed contribution. The manuscript needs to have a proper, scientifically correct discussion and more in-depth arguments for the style of the model test analysis of groundwater level fluctuations on karst cover deformation, including model validation, accuracy, and uncertainty (if any).

Response 5: Revise the conclusion as follows.

(1) Under the action of rainfall and other recharge, cavity supply and drainage, the infiltration curve of the soil in the cover layer changes faster at the beginning and slows down gradually at the later stage as the time and intensity of rainfall and other recharge increase, but the thicker the cover layer, the slower the overall deformation; when the rate of cavity supply and drainage is certain, the thicker the cover layer, the smaller the deformation produced by the fluctuation of the groundwater level; after the formation of the soil cavity, rainfall and other recharge, cavity After the formation of the cavity, rainfall and other recharge, cavity supply and drainage accelerate the deformation of the soil and the upward development of the cavity.

(2) Under the condition of uniform initial water content and density, the cumulative amount and time of rainfall and other recharge required for the deformation to collapse process increases as the thickness of the cover soil increases, and the number of cycles of cavity drainage required also increases. This indicates that the fluctuation frequency of groundwater level can aggravate the expansion of karst collapse.

(3) The karst cover is affected by a combination of self-gravity, subduction by surface water seepage, absorption by water-air change and airburst during the process of deformation to collapse. The process of deformation to collapse can be roughly divided into three stages: the first stage is the gradual softening of the soil from stability under the action of seepage subduction, the second stage is the slow deformation of the soil under the action of water-air change absorption and airburst, and the third stage is the overall rapid collapse of the soil under the combined action.

Round 2

Reviewer 2 Report

Authors did made changes in their manuscript based on reviewer comments. I feel that this lead to improvement of original manuscript; however, I still feel that this work has potential to be further improved as both topic and research are interesting. I leave final decision to the journal Editor. 

Author Response

Authors did made changes in their manuscript based on reviewer comments. I feel that this lead to improvement of original manuscript; however, I still feel that this work has potential to be further improved as both topic and research are interesting. I leave final decision to the journal Editor.

Thanks to your patient review and valuable comments, I have read the article again carefully and revised many of the details.The specific changes are marked in red in the article.

Reviewer 3 Report

The author is appreciated for making the corrections, but it seems that this reviewer's highlighted file in the first round did not reach the author correctly, or I did not upload the latest version correctly. Therefore, I request the author to address the other comments as well.

It is necessary to correct the following two sentences in the new abstract.

Line 11: ... the subduction process of groundwater, and the groundwater, ...

Line 12: ... physical model to model the ...

Author Response

Response to Reviewer 3 Comments

Point 1: Avoid long and misleading sentences. Rewrite the abstract by providing shorter and more understandable sentences.

Response 1:  I have broken up and revised the article with regard to the lengthy and misleading sentences in the abstract that you raised.

Abstract:Engineering practice and real-life cases show that the geological conditions of the Guilin overlying karst site are complex. In particular, the thickness of the overlying soil layer, which affects the speed of the subduction process of groundwater, and the groundwater, which drives the accelerated formation of soil cavities. Therefore, this paper is based on a physical model to model the effects of groundwater level changes caused by different factors on karst cover deformation. The model tests are simulated for different cover thicknesses (6cm, 9cm, 12cm, 15cm, 18cm) under rainfall and other recharge and cavity supply and drainage conditions for the same density (1.40 g/cm3) and initial water content (30%).The results show that with the increase of rainfall and other recharge time, the basic change trend of different cover thicknesses is that the infiltration curve changes faster at the beginning and slows down at the end, but the thicker the cover, the slower the overall deformation; at a certain rate of cavity recharge and drainage, the thicker the cover, the smaller the deformation caused by the fluctuation of groundwater level, and the cavity recharge makes the cover displacement obvious, in the order of 0.304cm, 0.173cm, 0.118cm, 0.068cm,and 0.056cm . After the formation of the cavity, the rainfall and other recharge, and the cavity supply and drainage accelerated the destruction and deformation of the soil body and the upward development of the cavity. The research results provide theoretical support for the subsequent prevention and control of karst collapse in covered karst areas, and have certain practical engineering significance.

Point 2: Always try to use new and state of the art references, especially in the introduction section. More than 60% of references should be chosen from the last 3-4 years. Unfortunately, in this manuscript, only less than 25% of references have such feature. Therefore, the introduction section should be rewritten.

Response 2: A description of the selected parameters has been added to the text, and the conclusions in the abstract have been recapitulated and revised.

Point 3: Avoid repeating sentences. These sentences are similar to the sentences in the final part of the abstract.

Response 3: In response to your suggestion to avoid repetitive sentences, and the sentences in lines 101-103 of my article are similar to those in the last part of the abstract. I have amended them.

Original text: The study provides provide theoretical support for the subsequent prevention and control of karst collapse in covered karst areas, and has certain practical engineering significance.

Modified: This study provides a theoretical and technical basis for practical engineering activities in covered karst areas.

Point 4: Additional explanations about the middle soil box are needed.

How were the upper tanks and soil separated from each other? What material is this boundary element made of? What is its permeability?

Response 4:The main model box has dimensions of L (120 cm) x W (70 cm) x H (100 cm) and is made up of two parts, the upper and lower, as shown in Figure 3. The upper part of this consists of a central soil box (100 cm x 70 cm x 60 cm) and water tanks (10 cm x 70 cm x 60 cm ) on the left and right sides.

The upper intermediate soil box is defined by the geological model generalisation and is therefore supplemented by the paragraph below 3.1 and 3.2. This is shown in additional detail below.

3.1.  Generalisation of the geological model of the study area

Due to the complex geological conditions in the study area, there are many uncertainties. If the karst-induced collapse test were to be carried out directly at the study site, it would be constrained by many factors. It is therefore difficult to carry out in-situ karst collapse tests at the site. In order to obtain the results of the karst collapse test under changing groundwater levels and to carry out the test successfully, a geological model of the geological conditions in the study area needs to be generalised here.

The geological drilling data collected from Lingui District, Guilin City, shows that the overlying karst in Lingui District can be mainly classified into single-layer structure or double-layer structure. Among the monolayer structures, the overlying soil layer is clayey. In this simulation test study, a specific study area was selected to focus on the monolithic structure (Figure 2).

Figure 2. Geological generalisation of a single structure

1—Overburden clayey soil;2—Limestone;3—Caves

3.2. Physical modelling principles

When conducting physical model tests, not only should the test requirements and operability be met, but similar issues should also be considered. According to the geological reports available in the study area, it is known that the thickness of the collapse prone overburden in the study area ranges from 0.7 to 5.6m, and the diameter of the karst openings on the bedrock surface varies from 0.5 to 1.2m. The geometric similarity ratio was finally determined to be 1:15 by combining the geological, hydrological and tectonic conditions of the study area. 6 cm diameter of the reserved soil hole was set, and the thickness of the overburden layer was 6 cm, 9 cm, 12 cm, 15 cm and 18 cm. The soil samples of the overburden layer were taken directly from the site during the test, and the moisture content of the test was close to that of the site. However, the density was difficult to achieve in situ, and after repeated indoor tests, the density was finally determined to be 1.4 g/cm³. In order to make the simulation conditions as close to the site as possible, to take into account the factors involved in karst collapse at the site and to ensure that the soil simulation parameters are basically the same as the original soil at the site, the karst collapse test device was specially tailored. In the course of the test, the test device is subjected to a large load when filling a predetermined thickness of overlying soil. This is why the frame of the device is made of a high hardness aluminium alloy and the side walls are inlaid with 5mm thick toughened glass. In order to have a clear view of the deformation and damage process inside the cover soil, the karst access opening was set up in a semi-circle at the midpoint of the front side. A special ice block was pre-buried above the passage opening to form a pre-existing soil cavity in the shape of a 1/4 sphere during the test.

The earth and water tanks are separated by 5mm thickness toughened glass. To accommodate the inflow of lateral recharge water during the test, holes were drilled in the glass on the left and right sides, as shown in Figure 1. To prevent loss of cover soil in the small holes on the left and right sides, towels were taped to the left and right sides before the test began to fill with soil, as shown in Figure 2.

Fig 1                                       Fig 2

Point 5: Please provide a higher quality photo and identify the various components on it.

Response 5: I have replaced the clear photos and added additional descriptions of the various parts.

1—Special rainfall shower                         2—Model unit side water tank

3—Overburden soil                               4—Existing earth cave

5—Dissolving chamber water tank                  6—Dissolution chamber inlet

7—Dissolution chamber drain                      8—Drainage monitoring water meters

9—Cameras

Point 6:Accuracy of monitoring equipment and data recording rate should be provided.

Response 6: For the accuracy of the monitoring equipment and the rate of data recording, I have added additional notes.

The deformation of the cover soil and the pressure of the karst cavity are monitored according to the test requirements of this paper. The displacement meter sensor model DMWY-100 is used for the determination of the cover soil deformation, with a compact size suitable for small-scale indoor model tests. The output is highly sensitive and can resolve displacement changes of less than 0.005mm; wide range, high accuracy, small drift and stable performance; stainless steel full bridge housing, corrosion resistant and pressure resistant; technical parameters are shown in Table 1.The pressure in the karst cavity is monitored using a levelogger 5 barometric pressure sensor to monitor changes in pressure during supply and drainage. A video camera is used to record the deformation to collapse of the overburden on the karst bedrock, with a focus on capturing the deformation, intensification and end of the karst collapse caused by rainfall and lateral recharge, water supply and drainage.

Table 1. Technical parameters of DMWY-100 surface type displacement meter

Dimensions

Φ25*345

Measuring range (mm)

0~100

Full output (μℇ)

8000

Correction factor (mm/μℇ)

0.0125

Precision

≤0.2%F•S 

Error (mm)

≤0.01

(Note:F•S is the full scale output value, around 8000 microstrain)

Point 7: The method of mixing the soil, filling the mold box and compacting it to reach the desired MDD should be clearly stated. How did you make sure that the soil after trimming has a density of 1.4 g/cm3?

Response 7: Firstly, based on the moisture content w0=5% measured after drying the soil samples taken from the study area, combined with the initial moisture content w1=30% and dry density pd=1.4g/cm3 determined in the test protocol, and the different cover thicknesses required for each test (6cm, 9cm, 12cm, 15cm, 18cm), the mass of soil m0 and the mass of water m2 required for each test were calculated respectively. Using a mulch layer thickness of 6cm as an example, the calculation process is as follows.

V=100×70×6

The calculated mass of soil m0 and the mass of water m2 are then configured to obtain the soil required for the test. Finally, the configured soil samples are filled in batches in the model box.As both scenarios 1 and 2 were the same soil structure in 5 different thicknesses, the filling protocol was the same. For the first 6 cm soil thickness test, the soil sample was divided into two parts and filled in two batches, with one sample filled to 3 cm and the remaining sample filled to 6 cm with surface trimming. Each subsequent test was filled to the required height for each test, following the same filling method as the first 6 cm.

Point 8:  The quality of the figure should be increased(Fig 2、Fig 3、Fig 5、Fig 7、Fig 8).

Response 8:In response to your questions about the line shape, thickness, clarity, etc. of the images in the article (Fig. 2, Fig. 3, Fig. 5, Fig. 7, Fig. 8, Fig. 9), I have made changes to each image.

Point 9: (1) It is suggested to present and analyze the method or pattern of collapse for different experiments (and due to different factors) using the images recorded by the camera during the tests.

(2)The results presented in this section should be validated by providing photographs of the tested samples.

(3)tests. Please consider this comment in the whole text.The mere description of what the authors observed in the laboratory is not acceptable at all. Any content expressed in the manuscript should be supported by providing relevant tables, figures or images extracted from the

Response 9: To address these 3 issues, I have added the relevant sections of section 4.3. This is shown below.

4.3. Analysis of the collapse development process

The surface water, pore water and groundwater in the lower part of the soil box are closely linked to form a complete hydrological cycle in the test model installation. The infiltration of surface water can change the water content of the cover soil. When rainfall or other recharge infiltrates, the water content of the cover soil increases, the cohesion and angle of internal friction decrease, the capacity and self-gravity also increase and the shear strength decreases, which ultimately leads to a reduction in collapse resistance. The effect of groundwater on karst ground collapse can be analysed in many ways, but the magnitude and frequency of groundwater fluctuations are the most important. Based on the development of the soil cavity under the two test scenarios the whole process can be roughly divided into a softening phase of the soil above the cavity, a soil loss-spalling phase and a rapid-integral collapse phase (Figure 12).

(a)Initial view of completed fill                     (b)Soil softening stage

(c)Soil erosion-exfoliation phase              (d)Rapid - overall collapse phase

 Figure 12. Schematic diagram of the development of the earth cave collapse

(1)Soil softening stage: At the beginning of the test, rainfall and lateral seepage recharge the groundwater and a gradient difference is formed between pore water and groundwater, resulting in vertical seepage. As the support for the cover soil above the soil hole is weak, it is susceptible to submerged erosion making the soil particles soften.

(2)Soil loss-exfoliation stage: With the increase of rainfall and lateral seepage recharge time, the saturation of the soil increases and the permeability becomes poor. When the water level rises, the gas inside the cavern cannot be discharged and is squeezed to form a high-pressure air mass, which produces tiny fissures in the cover soil. When the water level falls, the negative pressure inside the cavern increases the additional force of downward seepage of pore water in the cover soil, which intensifies the loss and spalling of the soil.

(3)Rapid - Integral Collapse Phase: Long periods of continuous recharge and fluctuations in the water table resulted in a large loss and spalling of the cover soil above the cavity in the later stages of the test, and the arch structure no longer existed. The cavity was observed earlier in the form of an altar shown in Figure 13, at which point the self-weight of the soil increased with recharge, while the thickness of the soil was still being lost and spalled with fluctuations in the water table. When the collapse develops to the point where the collapse causing force is greater than the collapse resisting force, the soil undergoes shear deformation falling off as a whole, eventually leading to the overall collapse of the surface, as shown in Figure 14.

    Figure 13. Cavity Altarpiece                   Figure 14. Surface collapse map

Point 10: Avoid repeating sentences. These sentences have already been stated.

Response 10: Lines 215-222 below Figure 3 in the text are from section 2.4 and were inadvertently added here as purely redundant, and I have removed them from the original text.

Point 11: Are the authors sure that this title is written correctly?

Response 11: Due to carelessness the content of my other article has been included in this paper and I have revised it.

Original title "Effect of contamination concentration on unconfined compressive strength"

Revised title "Analysis of the results of the dissolved cavity water level drop test "

Point 12: Conclusion

Response 12: The conclusions have been rewritten.

(2) Under the condition of uniform initial water content and density, the cumulative amount and time of rainfall and other recharge required for the deformation to collapse process increases as the thickness of the cover soil increases, and the number of cycles of cavity drainage required also increases. This indicates that the fluctuation frequency of groundwater level can aggravate the expansion of karst collapse.

(3) The karst cover is affected by a combination of self-gravity, subduction by surface water seepage, suction by water-air change and airburst during the d

Response to Reviewer 3 Comments

Point 1: Avoid long and misleading sentences. Rewrite the abstract by providing shorter and more understandable sentences.

Response 1:  I have broken up and revised the article with regard to the lengthy and misleading sentences in the abstract that you raised.

Abstract:Engineering practice and real-life cases show that the geological conditions of the Guilin overlying karst site are complex. In particular, the thickness of the overlying soil layer, which affects the speed of the subduction process of groundwater, and the groundwater, which drives the accelerated formation of soil cavities. Therefore, this paper is based on a physical model to model the effects of groundwater level changes caused by different factors on karst cover deformation. The model tests are simulated for different cover thicknesses (6cm, 9cm, 12cm, 15cm, 18cm) under rainfall and other recharge and cavity supply and drainage conditions for the same density (1.40 g/cm3) and initial water content (30%).The results show that with the increase of rainfall and other recharge time, the basic change trend of different cover thicknesses is that the infiltration curve changes faster at the beginning and slows down at the end, but the thicker the cover, the slower the overall deformation; at a certain rate of cavity recharge and drainage, the thicker the cover, the smaller the deformation caused by the fluctuation of groundwater level, and the cavity recharge makes the cover displacement obvious, in the order of 0.304cm, 0.173cm, 0.118cm, 0.068cm,and 0.056cm . After the formation of the cavity, the rainfall and other recharge, and the cavity supply and drainage accelerated the destruction and deformation of the soil body and the upward development of the cavity. The research results provide theoretical support for the subsequent prevention and control of karst collapse in covered karst areas, and have certain practical engineering significance.

Point 2: Always try to use new and state of the art references, especially in the introduction section. More than 60% of references should be chosen from the last 3-4 years. Unfortunately, in this manuscript, only less than 25% of references have such feature. Therefore, the introduction section should be rewritten.

Response 2: A description of the selected parameters has been added to the text, and the conclusions in the abstract have been recapitulated and revised.

Point 3: Avoid repeating sentences. These sentences are similar to the sentences in the final part of the abstract.

Response 3: In response to your suggestion to avoid repetitive sentences, and the sentences in lines 101-103 of my article are similar to those in the last part of the abstract. I have amended them.

Original text: The study provides provide theoretical support for the subsequent prevention and control of karst collapse in covered karst areas, and has certain practical engineering significance.

Modified: This study provides a theoretical and technical basis for practical engineering activities in covered karst areas.

Point 4: Additional explanations about the middle soil box are needed.

How were the upper tanks and soil separated from each other? What material is this boundary element made of? What is its permeability?

Response 4:The main model box has dimensions of L (120 cm) x W (70 cm) x H (100 cm) and is made up of two parts, the upper and lower, as shown in Figure 3. The upper part of this consists of a central soil box (100 cm x 70 cm x 60 cm) and water tanks (10 cm x 70 cm x 60 cm ) on the left and right sides.

The upper intermediate soil box is defined by the geological model generalisation and is therefore supplemented by the paragraph below 3.1 and 3.2. This is shown in additional detail below.

3.1.  Generalisation of the geological model of the study area

Due to the complex geological conditions in the study area, there are many uncertainties. If the karst-induced collapse test were to be carried out directly at the study site, it would be constrained by many factors. It is therefore difficult to carry out in-situ karst collapse tests at the site. In order to obtain the results of the karst collapse test under changing groundwater levels and to carry out the test successfully, a geological model of the geological conditions in the study area needs to be generalised here.

The geological drilling data collected from Lingui District, Guilin City, shows that the overlying karst in Lingui District can be mainly classified into single-layer structure or double-layer structure. Among the monolayer structures, the overlying soil layer is clayey. In this simulation test study, a specific study area was selected to focus on the monolithic structure (Figure 2).

Figure 2. Geological generalisation of a single structure

1—Overburden clayey soil;2—Limestone;3—Caves

3.2. Physical modelling principles

When conducting physical model tests, not only should the test requirements and operability be met, but similar issues should also be considered. According to the geological reports available in the study area, it is known that the thickness of the collapse prone overburden in the study area ranges from 0.7 to 5.6m, and the diameter of the karst openings on the bedrock surface varies from 0.5 to 1.2m. The geometric similarity ratio was finally determined to be 1:15 by combining the geological, hydrological and tectonic conditions of the study area. 6 cm diameter of the reserved soil hole was set, and the thickness of the overburden layer was 6 cm, 9 cm, 12 cm, 15 cm and 18 cm. The soil samples of the overburden layer were taken directly from the site during the test, and the moisture content of the test was close to that of the site. However, the density was difficult to achieve in situ, and after repeated indoor tests, the density was finally determined to be 1.4 g/cm³. In order to make the simulation conditions as close to the site as possible, to take into account the factors involved in karst collapse at the site and to ensure that the soil simulation parameters are basically the same as the original soil at the site, the karst collapse test device was specially tailored. In the course of the test, the test device is subjected to a large load when filling a predetermined thickness of overlying soil. This is why the frame of the device is made of a high hardness aluminium alloy and the side walls are inlaid with 5mm thick toughened glass. In order to have a clear view of the deformation and damage process inside the cover soil, the karst access opening was set up in a semi-circle at the midpoint of the front side. A special ice block was pre-buried above the passage opening to form a pre-existing soil cavity in the shape of a 1/4 sphere during the test.

The earth and water tanks are separated by 5mm thickness toughened glass. To accommodate the inflow of lateral recharge water during the test, holes were drilled in the glass on the left and right sides, as shown in Figure 1. To prevent loss of cover soil in the small holes on the left and right sides, towels were taped to the left and right sides before the test began to fill with soil, as shown in Figure 2.

Fig 1                                       Fig 2

Point 5: Please provide a higher quality photo and identify the various components on it.

Response 5: I have replaced the clear photos and added additional descriptions of the various parts.

1—Special rainfall shower                         2—Model unit side water tank

3—Overburden soil                               4—Existing earth cave

5—Dissolving chamber water tank                  6—Dissolution chamber inlet

7—Dissolution chamber drain                      8—Drainage monitoring water meters

9—Cameras

Point 6:Accuracy of monitoring equipment and data recording rate should be provided.

Response 6: For the accuracy of the monitoring equipment and the rate of data recording, I have added additional notes.

The deformation of the cover soil and the pressure of the karst cavity are monitored according to the test requirements of this paper. The displacement meter sensor model DMWY-100 is used for the determination of the cover soil deformation, with a compact size suitable for small-scale indoor model tests. The output is highly sensitive and can resolve displacement changes of less than 0.005mm; wide range, high accuracy, small drift and stable performance; stainless steel full bridge housing, corrosion resistant and pressure resistant; technical parameters are shown in Table 1.The pressure in the karst cavity is monitored using a levelogger 5 barometric pressure sensor to monitor changes in pressure during supply and drainage. A video camera is used to record the deformation to collapse of the overburden on the karst bedrock, with a focus on capturing the deformation, intensification and end of the karst collapse caused by rainfall and lateral recharge, water supply and drainage.

Table 1. Technical parameters of DMWY-100 surface type displacement meter

Dimensions

Φ25*345

Measuring range (mm)

0~100

Full output (μℇ)

8000

Correction factor (mm/μℇ)

0.0125

Precision

≤0.2%F•S 

Error (mm)

≤0.01

(Note:F•S is the full scale output value, around 8000 microstrain)

Point 7: The method of mixing the soil, filling the mold box and compacting it to reach the desired MDD should be clearly stated. How did you make sure that the soil after trimming has a density of 1.4 g/cm3?

Response 7: Firstly, based on the moisture content w0=5% measured after drying the soil samples taken from the study area, combined with the initial moisture content w1=30% and dry density pd=1.4g/cm3 determined in the test protocol, and the different cover thicknesses required for each test (6cm, 9cm, 12cm, 15cm, 18cm), the mass of soil m0 and the mass of water m2 required for each test were calculated respectively. Using a mulch layer thickness of 6cm as an example, the calculation process is as follows.

V=100×70×6

The calculated mass of soil m0 and the mass of water m2 are then configured to obtain the soil required for the test. Finally, the configured soil samples are filled in batches in the model box.As both scenarios 1 and 2 were the same soil structure in 5 different thicknesses, the filling protocol was the same. For the first 6 cm soil thickness test, the soil sample was divided into two parts and filled in two batches, with one sample filled to 3 cm and the remaining sample filled to 6 cm with surface trimming. Each subsequent test was filled to the required height for each test, following the same filling method as the first 6 cm.

Point 8:  The quality of the figure should be increased(Fig 2、Fig 3、Fig 5、Fig 7、Fig 8).

Response 8:In response to your questions about the line shape, thickness, clarity, etc. of the images in the article (Fig. 2, Fig. 3, Fig. 5, Fig. 7, Fig. 8, Fig. 9), I have made changes to each image.

Point 9: (1) It is suggested to present and analyze the method or pattern of collapse for different experiments (and due to different factors) using the images recorded by the camera during the tests.

(2)The results presented in this section should be validated by providing photographs of the tested samples.

(3)tests. Please consider this comment in the whole text.The mere description of what the authors observed in the laboratory is not acceptable at all. Any content expressed in the manuscript should be supported by providing relevant tables, figures or images extracted from the

Response 9: To address these 3 issues, I have added the relevant sections of section 4.3. This is shown below.

4.3. Analysis of the collapse development process

The surface water, pore water and groundwater in the lower part of the soil box are closely linked to form a complete hydrological cycle in the test model installation. The infiltration of surface water can change the water content of the cover soil. When rainfall or other recharge infiltrates, the water content of the cover soil increases, the cohesion and angle of internal friction decrease, the capacity and self-gravity also increase and the shear strength decreases, which ultimately leads to a reduction in collapse resistance. The effect of groundwater on karst ground collapse can be analysed in many ways, but the magnitude and frequency of groundwater fluctuations are the most important. Based on the development of the soil cavity under the two test scenarios the whole process can be roughly divided into a softening phase of the soil above the cavity, a soil loss-spalling phase and a rapid-integral collapse phase (Figure 12).

(a)Initial view of completed fill                     (b)Soil softening stage

(c)Soil erosion-exfoliation phase              (d)Rapid - overall collapse phase

 Figure 12. Schematic diagram of the development of the earth cave collapse

(1)Soil softening stage: At the beginning of the test, rainfall and lateral seepage recharge the groundwater and a gradient difference is formed between pore water and groundwater, resulting in vertical seepage. As the support for the cover soil above the soil hole is weak, it is susceptible to submerged erosion making the soil particles soften.

(2)Soil loss-exfoliation stage: With the increase of rainfall and lateral seepage recharge time, the saturation of the soil increases and the permeability becomes poor. When the water level rises, the gas inside the cavern cannot be discharged and is squeezed to form a high-pressure air mass, which produces tiny fissures in the cover soil. When the water level falls, the negative pressure inside the cavern increases the additional force of downward seepage of pore water in the cover soil, which intensifies the loss and spalling of the soil.

(3)Rapid - Integral Collapse Phase: Long periods of continuous recharge and fluctuations in the water table resulted in a large loss and spalling of the cover soil above the cavity in the later stages of the test, and the arch structure no longer existed. The cavity was observed earlier in the form of an altar shown in Figure 13, at which point the self-weight of the soil increased with recharge, while the thickness of the soil was still being lost and spalled with fluctuations in the water table. When the collapse develops to the point where the collapse causing force is greater than the collapse resisting force, the soil undergoes shear deformation falling off as a whole, eventually leading to the overall collapse of the surface, as shown in Figure 14.

    Figure 13. Cavity Altarpiece                   Figure 14. Surface collapse map

Point 10: Avoid repeating sentences. These sentences have already been stated.

Response 10: Lines 215-222 below Figure 3 in the text are from section 2.4 and were inadvertently added here as purely redundant, and I have removed them from the original text.

Point 11: Are the authors sure that this title is written correctly?

Response 11: Due to carelessness the content of my other article has been included in this paper and I have revised it.

Original title "Effect of contamination concentration on unconfined compressive strength"

Revised title "Analysis of the results of the dissolved cavity water level drop test "

Point 12: Conclusion

Response 12: The conclusions have been rewritten.

(2) Under the condition of uniform initial water content and density, the cumulative amount and time of rainfall and other recharge required for the deformation to collapse process increases as the thickness of the cover soil increases, and the number of cycles of cavity drainage required also increases. This indicates that the fluctuation frequency of groundwater level can aggravate the expansion of karst collapse.

(3) The karst cover is affected by a combination of self-gravity, subduction by surface water seepage, suction by water-air change and airburst during the deformation to collapse process. For the same thickness of cover soil, the relationship between the deformation due to different effects is: water level rise is greater than water level fall than rainfall and lateral recharge, for example, with a cover of 6 cm: water level rise (3.04 mm) > water level fall (1.63 mm) > rainfall and lateral recharge (1.44 mm).

eformation to collapse process. For the same thickness of cover soil, the relationship between the deformation due to different effects is: water level rise is greater than water level fall than rainfall and lateral recharge, for example, with a cover of 6 cm: water level rise (3.04 mm) > water level fall (1.63 mm) > rainfall and lateral recharge (1.44 mm).

Reviewer 4 Report

Based on the revised manuscript, I still strongly believe that the manuscript is not acceptable yet for the Applied Sciences journal. The improvements made by the authors are not profound and insufficient to make this paper a scientific contribution. The manuscript still has no real discussion yet (we cannot see a proper and scientifically correct discussion), and also, where interpretations are expected, lacks detailed referencing to demonstrate that the author tries to put their results in context. I am concerned that the results of this study will not contribute to research and practice.

Author Response

Based on the revised manuscript, I still strongly believe that the manuscript is not acceptable yet for the Applied Sciences journal. The improvements made by the authors are not profound and insufficient to make this paper a scientific contribution. The manuscript still has no real discussion yet (we cannot see a proper and scientifically correct discussion), and also, where interpretations are expected, lacks detailed referencing to demonstrate that the author tries to put their results in context. I am concerned that the results of this study will not contribute to research and practice.

I have made changes to the logical framework and details throughout the article, and the additions and changes are marked in red in the original text.

Round 3

Reviewer 3 Report

The authors are appreciated for addressing the comments. Currently, the quality of the manuscript has improved to a good extent, but some of the comments from the previous round have been ignored and need to be corrected. For example, some of the ignored comments are:

1. Lines 10-12: Change to: In particular, the groundwater, which drives the accelerated formation of soil cavities, and the thickness of the overlying soil layer, which affects the speed of the groundwater subsidence process.

2. Lines 12-13: “physical model to model the effects” change to “physical model to evaluate the effects”

3. It is suggested that the following paragraph provided in the response file be added to the manuscript. “The earth and water tanks are separated by 5mm thickness toughened glass. To accommodate the inflow of lateral recharge water during the test, holes were drilled in the glass on the left and right sides, as shown in Figure 1. To prevent loss of cover soil in the small holes on the left and right sides, towels were taped to the left and right sides before the test began to fill with soil, as shown in Figure 2.”

4. It is recommended to add Figures 1 and 2 provided in the response file to the manuscript (these two figures can be merged with Figure 3 in the manuscript).

5. The sentences and equations presented in lines 239 to 247 can be deleted.

6. The quality of the Figure 4 should be increased.

7. In the previous round, the authors were asked to provide credible references for some of the claims made in the section of "Analysis of test", which was unfortunately ignored.

8. The titles of sections 4.1 and 4.2 are the same!!!! The authors are strongly advised to double-check the titles and numbering of headings and subheadings.

Line 308: “period. .“ The extra dot must be deleted.

Line 307: “0.This” change to “0. This”

Author Response

 Lines 10-12: Change to: In particular, the groundwater, which drives the accelerated formation of soil cavities, and the thickness of the overlying soil layer, which affects the speed of the groundwater subsidence process.

For lines 10-12, I have amended them in line with your comments.

Reviewer 4 Report

The author has tried to make improvements according to the reviewer's suggestions and the recent manuscript is way better than the first version. However, some references for their statements in the manuscript or what they wrote are still absent/missing, regarding that specific information, it should be an article or book they used. For example, lines 101-131.

The most important thing is to compare their results in the test analysis (lines 277-438) with other research. For example, the author explained the effect of different recharge on cover deformation. They need to compare their results with another article (either from model or fieldwork measurements), so the readers can understand better. From the model, there is an effect but in reality, is there any effect of different recharge on cover deformation? Likewise with other sub-topics, i.e., the impact of groundwater level fluctuation and analysis of the collapse development process. We see now that the authors did a monologue about their results (no references at all) but forgot to discuss their results with other research. 

Author Response

 The author has tried to make improvements according to the reviewer's suggestions and the recent manuscript is way better than the first version. However, some references for their statements in the manuscript or what they wrote are still absent/missing, regarding that specific information, it should be an article or book they used. For example, lines 101-131.

  The content written in lines 101-131 is the geological setting of the study area. The two main aspects are meteorology and hydrology, and geological features. The meteorological and hydrological information is taken from the reports of the local meteorological office, and the geological features are taken from exploration information from actual engineering cases in the study area. No specific articles or books are available.
